METHODS

# Testing for association with rare variants in the coding and non-coding genome: RAVA-FIRST, a new approach based on CADD deleteriousness score

Ozvan Bocher[1,2]*, Thomas E. Ludwig[1,3], Marie-Sophie Oglobinsky[1], Gaëlle Marenne[1], Jean-François Deleuze[4], Suryakant Suryakant[5], Jacob Odeberg[6,7], Pierre-Emmanuel Morange[8], David-Alexandre Trégouët[5], Hervé Perdry[9], Emmanuelle Génin[1,3]

**1** Univ Brest, Inserm, EFS, UMR 1078, GGB, Brest, France, **2** Institute of Translational Genomics, Helmholtz Zentrum München, Munich, Germany, **3** CHU Brest, Brest, France, **4** Centre National de Recherche en Génomique Humaine CNRGH, Institut de Biologie François Jacob, Université Paris Saclay, CEA, Evry, France, **5** University of Bordeaux, Inserm, Bordeaux Population Health Research Center, team ELEANOR, UMR 1219, Bordeaux, France, **6** Science for Life Laboratory, Department of Protein Science, CBH, KTH Royal Institute of Technology, Stockholm, Sweden, **7** Department of Clinical Medicine, Faculty of Health Science, The Arctic University of Tromsö, Tromsö, Norway, **8** Aix Marseille Univ, INSERM, INRAE, C2VN, Marseille, France, **9** CESP Inserm, U1018, UFR Médecine, Univ Paris-Sud, Université Paris-Saclay, Villejuif, France

\* bocherozvan@gmail.com

**Data Availability Statement:** The files containing the positions of CADD regions, the positions of genomic categories and the adjusted CADD scores

## Abstract

Rare variant association tests (RVAT) have been developed to study the contribution of rare variants widely accessible through high-throughput sequencing technologies. RVAT require to aggregate rare variants in testing units and to filter variants to retain only the most likely causal ones. In the exome, genes are natural testing units and variants are usually filtered based on their functional consequences. However, when dealing with whole-genome sequence (WGS) data, both steps are challenging. No natural biological unit is available for aggregating rare variants. Sliding windows procedures have been proposed to circumvent this difficulty, however they are blind to biological information and result in a large number of tests. We propose a new strategy to perform RVAT on WGS data: "RAVA-FIRST" (RAre Variant Association using Functionally-InfoRmed STeps) comprising three steps. (1) New testing units are defined genome-wide based on functionally-adjusted Combined Annotation Dependent Depletion (CADD) scores of variants observed in the gnomAD populations, which are referred to as "CADD regions". (2) A region-dependent filtering of rare variants is applied in each CADD region. (3) A functionally-informed burden test is performed with sub-scores computed for each genomic category within each CADD region. Both on simulations and real data, RAVA-FIRST was found to outperform other WGS-based RVAT. Applied to a WGS dataset of venous thromboembolism patients, we identified an intergenic region on chromosome 18 enriched for rare variants in early-onset patients. This region that was missed by standard sliding windows procedures is included in a TAD region that contains a

are available at https://lysine.univ-brest.fr/RAVA-FIRST/. All the functions needed for RAVA-FIRST to annotate, group, filter and analyse rare variants have been implemented in the R package Ravages (https://cran.r-project.org/web/packages/Ravages/, https://github.com/genostats/Ravages) which directly downloads the files from https://lysine.univ-brest.fr/RAVA-FIRST/. Information about the CADD region R126442 that was found associated with VTE age at first event is available in the S3 File. Information about individuals (WSS score, age and sex) and variants (position, adjusted CADD score and weight in WSS) are given. Data from the MARTHA study contain potentially identifying information. They are only available on reasonable request from the Biological Resource Center of the Assistance-Publique Hôpitaux de Marseille (AP-HM). For inquiries, please contact CRB.Timone. Marseille@ap-hm.fr.

**Funding:** This project was carried out in the framework of the INSERM GOLD Cross-Cutting program and was supported by the GENMED Laboratory of Excellence on Medical Genomics [ANR-10-LABX-0013], a research program managed by the National Research Agency (ANR) as part of the French Investment for the Future. DA. T was supported by the «EPIDEMIOM-VTE» Senior Chair from the Initiative of Excellence of the University of Bordeaux. The proteomics screening in MARTHA was financed by a grant from Stockholm County Council (SLL 2017-0842) and from Familjen Erling Perssons Foundation. The funders had no role in study design, data collection and analysis, decision to publish, or preparation of the manuscript.

**Competing interests:** The authors have declared that no competing interests exist.

strong candidate gene. RAVA-FIRST enables new investigations of rare non-coding variants in complex diseases, facilitated by its implementation in the R package Ravages.

## Author summary

Technological progresses have made possible whole-genome sequencing at an unprecedented scale, opening up the possibility to explore the role of genetic variants of low frequency in common diseases. The challenge is now methodological and requires the development of novel methods and strategies to analyse sequencing data that are not limited to assessing the role of coding variants. With RAVA-FIRST, we propose a novel strategy to investigate the role of rare variants in the whole-genome that takes benefit from biological information. Especially, RAVA-FIRST relies on testing units that go beyond genes to gather rare variants in the association tests. In this work, we show that this new strategy presents several advantages compared to existing methods. RAVA-FIRST offers an easy and straightforward analysis of genome-wide rare variants, especially the intergenic ones which are frequently left behind, making it a promising tool to get a better understanding of the biology of complex diseases.

## Introduction

With advance in sequencing technologies, it is now possible to explore the role of rare genetic variants in complex diseases. Rare variant association tests (RVAT) have been developed that gather rare variants into testing units and compare their rare variant content between cases and controls [1–3]. While the impact of rare variants has already been shown in several complex diseases [4–6], RVAT face two key challenges: (i) the definition of the testing units and (ii) the selection of the qualifying rare variants to include in these units. The proportion of causal variants in the testing units being a major driver of power, especially for burden tests, it is indeed important to ensure that qualifying variants are enriched in variants likely to have some functional impact [3, 7]. When exome analyses are undertaken, rare variants are most often grouped by genes and included in the analysis depending on their impact on the corresponding protein [8, 9]. Nevertheless, the gene definition is not always optimal as differences in rare variants burden between cases and controls could sometimes only be found in a sub-region of a gene. This is for example the case in the *RNF213* gene where an enrichment in rare variants located in the C-terminal region was found in Moyamoya cases [10]. Defining testing units and qualifying variants is much more challenging in the non-coding genome due to the lack of defined genomic elements and the higher difficulty to predict the functional impact of non-coding variants [11]. It is yet a question of interest as several studies have shown the importance of rare non-coding variants in the development of complex diseases [12–14]. Functional elements such as enhancers or promoters can be used as testing units [5,15,16]. However, these elements only cover a small portion of the non-coding genome and their size is often too small to gather a sufficient number of rare variants. On the other hand, sliding windows procedures such as SCAN-G [17] or WGSCAN [18] can be used to test for association over the whole-genome. Nevertheless, they present several limits including the window definition that is arbitrary and blind to biological information, the high number of tests and the associated computation time. With overlapping windows, there is also a strong correlation between the different testing units that requires permutation procedures to account for

multiple testing. Finally, to filter rare variants in the testing units, pathogenicity scores are often used but without guidelines on which score to use and which threshold to apply.

In this paper, we propose RAVA-FIRST (RAre Variant Association using Functionally InfoRmed STeps), a new strategy for analysing rare variants in the coding and the non-coding genome that addresses the previous issues. First, we provide pre-defined testing units in the whole genome called "CADD regions" based on the Combined Annotation Dependent Depletion (CADD) scores of deleteriousness of variants observed in the gnomAD general population. Second, we propose a filtering approach based on CADD scores with region-dependant thresholds to represent the genetic context of each CADD region and avoid the use of a fixed threshold along the genome. Finally, we integrate functional information into the burden test to detect an accumulation of rare variants in specific genomic categories within CADD regions. Through a statistical description of these CADD regions, we show that they preserve the integrity of the majority of functional elements in the genome. We also show that the RAVA-FIRST filtering strategy enables a better discrimination between functional and non-functional variants. We applied RAVA-FIRST to real whole-genome sequencing data from individuals with venous thromboembolism (VTE) and detected an intergenic association signal that would have been missed with sliding windows and a classical filtering of rare variants. RAVA-FIRST is implemented in the R package Ravages available on the CRAN and maintained on Github [19,20].

## Description of the method

### Ethics statement

The MARTHA study was approved by its institutional ethics committee and informed written consent was obtained in accordance with the Declaration of Helsinki. Ethics approval were obtained from the "Departement santé de la direction générale de la recherche et de l'innovation du ministère" (Projects DC: 2008–880 and 09.576).

RAVA-FIRST is developed to test for association with rare variants in the whole genome. It deals with all steps from the definition of testing units and the filtering of rare variants, to the association test accounting for functional information. The main steps are described here and represented in Fig 1 and further details are provided in S1 File and S1 Fig.

### Testing units in RAVA-FIRST: The CADD regions

To define testing units for association tests, we took inspiration from the work of Havrilla et al. (2019) [21]. They defined "constrained coding regions" (CCR) as exonic regions where no important functional variation (defined as being at least missense) was found in the general population of gnomAD [22]. Those regions could be of interest in RVAT as we can expect that an accumulation of rare variants within them would lead to an increased risk of developing a disease. However, in our experience, two limits prevent the direct use of CCR as testing units in the whole genome: they are too small to gather a sufficient number of rare variants (224 bp being the maximum length of a CCR) and their definition relies on the consequence of the variants on the translated protein, not available in the non-coding genome. To define regions in the non-coding genome using the same underlying hypothesis, we therefore decided to expand the proposed approach by estimating the functionality of variants through CADD scores [23]. CADD scores were chosen because of their availability for every substitution in the genome and because they rank well in the comparison test of functional annotation tools [24]. The goal here is to split the genome into regions according to the distribution of functional variation observed in gnomAD and not to detect the most constrained regions as aimed by Havrilla et al (2019) [21].

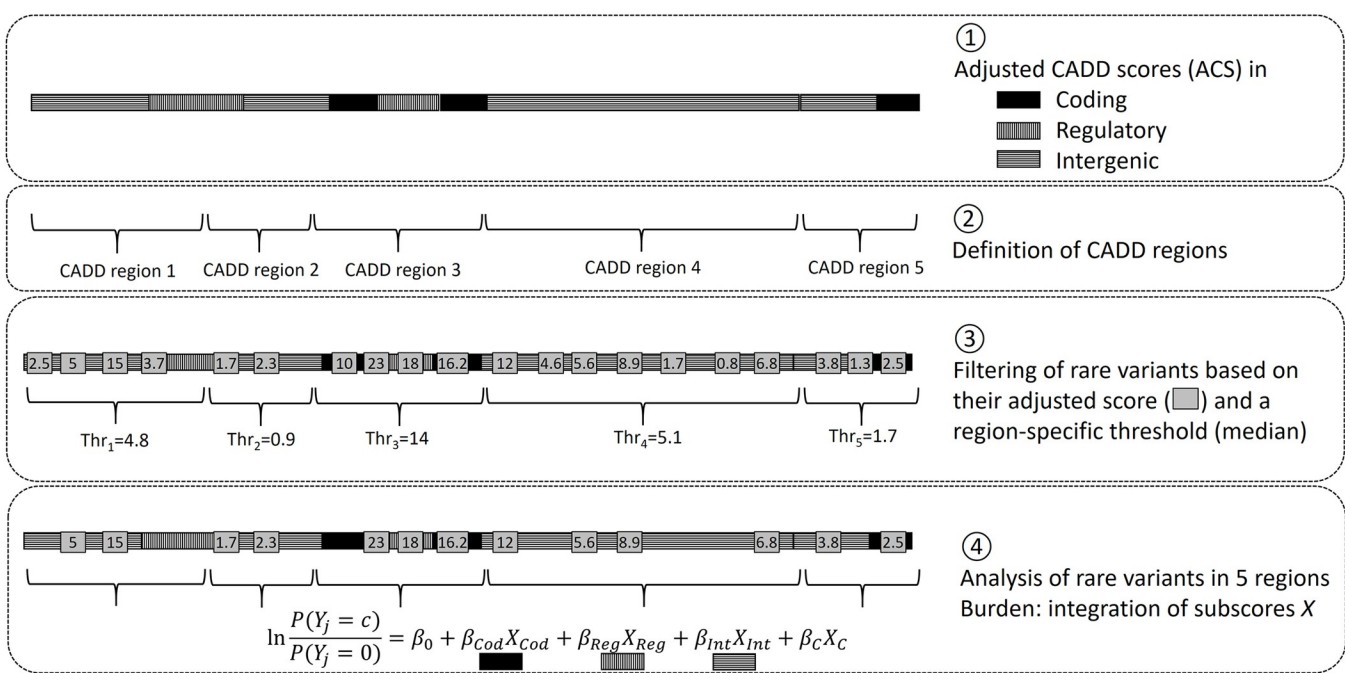

**Fig 1. Steps performed in RAVA-FIRST: definition of ACS, CADD regions, region-specific thresholds and functionally-informed burden tests.**

Coding variants tend to present higher CADD values than non-coding variants [23]. A selection based on a CADD threshold would therefore result in a majority of coding variants selected. In order to avoid this pattern, we adjusted the RAW CADD scores of all possible SNVs and of a set of 48,000,000 Indels on a PHRED scale within each of three genomic categories: "coding", "regulatory" and "intergenic" regions to obtain an "adjusted CADD score" also called "ACS". Coding regions correspond to CCDS [25] and represent 1.2% of the genome. Regulatory regions represent 44.3% of the genome and gather introns, 5' and 3' UTR, promoters and enhancers, all being involved in gene regulation [26]. Enhancers and promoters have been obtained with the SCREEN tool from ENCODE which enables the definition of a large number of regulatory elements in diverse cell types [27]. Finally, intergenic regions correspond to all other regions and represent 54.5% of the genome. More details are given in the S1 File.

ACS were used to select the variants that will bound the "CADD regions" based on criteria defined from a fine tuning to ensure that CADD regions had sizes compatible with RVAT; i.e., not too small to contain enough rare causal variants in cases and not too large to avoid pollution by too many rare neutral variants. First, we selected the variants with an ACS greater than 20, which is the top 1% of variants with the highest predicted functional impact within each of the three genomic categories. Then, among those variants, only the ones observed at least two times in gnomAD r2.0.1 genomes were used as boundaries of CADD regions. The choice of excluding gnomAD singletons was made to avoid splitting CADD regions because of sequencing errors. Contiguous small regions of less than 10 kb were grouped together. All genomic regions where CADD scores are not available (such as centromeres and telomeres among others) were excluded, as well as regions that are not sequenced or are low-covered in gnomAD but contain genomic sites where predicted ACS exceeds 20 for at least one of the possible alleles. This creates gaps within CADD regions that are sometimes of only one base pair but avoids keeping artificially long CADD regions due to a lack of observed variants in gnomAD.

More details about the steps and parameters used for the definition of CADD regions are presented in the S1 File.

## The RAVA-FIRST filtering strategy

In addition to the definition of new testing units in the whole genome, we propose a new filtering strategy in RAVA-FIRST to select qualifying variants based on thresholds that are specific to each CADD region. The idea is similar to the gene-specific CADD thresholds proposed by Itan et al (27) to improve variant deleteriousness prediction. To define region-specific thresholds, we computed the median of ACS of all the variants (SNPs and InDels) observed at least two times in gnomAD in each CADD region. This value is expected to represent the median score level that is tolerated in the general population within each CADD region. Qualifying variants are then defined as rare variants with an ACS above the threshold specific to their region. We chose to include InDels in this median so that they can be analysed using the RAVA-FIRST strategy as they represent an important source of genetic variation.

## Burden test in RAVA-FIRST: Taking into account functional information

Several of the CADD regions overlap different genomic categories (coding, regulatory or intergenic, Figs 1 and S2). As the effects of variants belonging to these different genomic categories may not be the same, we extended the burden test by integrating a sub-score for each genomic category into the regression model, similarly to the analysis of rare and frequent variants proposed by Li and Leal (2008) [7]:

$$\ln \frac{P(Y_j = 1)}{P(Y_j = 0)} = \beta_0 + \beta_{Cov}X_{Cov} + \sum_{G=\{cod;reg;inter\}} \beta_G X_G$$

$Y_j$ is the vector of phenotypes for the n individuals: 0 for the group of controls and 1 for the group of cases. $\beta_0$ represents the intercept of the model and $X_{Cov}$ the matrix of covariates (if any) with their associated effect, $\beta_{Cov}$. $\beta_G$ corresponds to the estimated effect of the burden $X_G$ within each genomic category within the tested CADD region. It can be computed for example using WSS [1], which corresponds to a weighted sum of rare alleles based on their frequency, the rarest alleles having the highest weights.

Sub-scores $X_G$ are thus constructed for each genomic category within a CADD region, with at most three sub-scores (coding, regulatory or intergenic). The p-value can be determined using a likelihood ratio test comparing this model to the null model where the sub-scores are not included. This sub-score analysis, referred to as RAVA-FIRST burden test, is also available for continuous and for categorical phenotypes using the extension of burden tests developed in Bocher et al. (2019) [19]. The RAVA-FIRST burden test coupled with the region-specific filtering on the ACS enables to perform only one test by CADD region while keeping the most important functional variants within each genomic category and accounting for those categories in the association test.

## Verification and comparison

### Statistics on CADD regions and comparison with genomic elements

A total of 135,224 CADD regions were defined covering 93.2% of the genome (in build GRCh37), of which 106,251 CADD regions are larger than 1kb (covering 93% of the genome). Overall, 42.1% of CADD regions span only one type of genomic category, 47.5% span two of the three types of genomic categories, and 10.4% overlap the three genomic categories (S2 Fig). Some CADD regions are extremely large, mainly in the regions close to the centromeres

**Table 1. Summary statistics of the lengths of CADD regions.**

| | Quantiles | | | | | Mean |
|---|---|---|---|---|---|---|
| | 0% | 25% | 50% | 75% | 100% | |
| Length (kb) | 0.002 | 2.576 | 13.006 | 24.323 | 1,731.228 | 19.852 |

(Table 1). Care should be taken when interpreting results obtained in these regions. Indeed, only few high-quality genomes covering these genomic regions are currently available and CADD scores may not be as reliable as in other parts of the genome. About 70% of CADD regions have a size between 1 and 50 kb with a mean of 20 kb, making them completely compatible with the size of genes commonly used as testing units in RVAT.

We then compared the position of genomic elements relative to the defined CADD regions (see S1 Table for the definition of genomic elements). A large majority of genomic elements are entirely included into a single CADD region and thus their integrity is preserved (see S2 Table). This is expected as all these genomic elements are substantially smaller than the CADD. For larger elements such as introns or lncRNA, the percentage decreases but remains high (more than 80% of lncRNA are overlapped by at most two CADD regions). The genomic elements spanning more than one CADD region are on average longer than the ones being entirely included into a single CADD region. However, when comparing CCR and CADD regions, it is interesting to note that the CCRs entirely encompassed within a single CADD region are the longest ones that also represent the most constrained regions.

## Performance of RAVA-FIRST filtering based on ACS

To assess the performance of the ACS and the RAVA-FIRST filtering, we evaluated its capacity in discriminating rare pathogenic SNVs defined in the Clinvar database [28] from rare SNVs polymorphisms observed in the 1000Genomes project [29]. We computed true positive rate (TPR), true negative rate (TNR) and precision for the RAVA-FIRST filtering and compared the results to the ones obtained by applying a fixed CADD threshold of 10, 15 or 20 on variants annotated with CADD scores v1.4. A total of 82,811 variants (44,566 benign and 38,245 pathogenic), both coding and non-coding, were included in this analysis (see S1 and S2 Files for more details on the selection of variants).

For coding variants, all filtering strategies based on CADD scores (fixed threshold or ACS) show a very high TPR (Fig 2A), meaning that the majority of pathogenic variants would be selected as qualifying variants for RVAT. The TNR increases with the increasing CADD score threshold which is expected as less variants, and therefore less benign variants, are included in the analysis. The RAVA-FIRST filtering shows the highest TNR and the highest precision. While the TPR value is extremely important to select the most probable causal variants in RVAT, it is also important to have a high TNR value, otherwise the signal will be diluted by a high proportion of non-causal variants. The precision value summarises the TPR and TNR parameters and is representative, to a certain extent, of the percentage of causal variants among selected variants. Therefore, we show that the RAVA-FIRST filtering strategy is the most accurate to select qualifying rare variants for RVAT. Focusing on the coding genome, we also compared the performance of RAVA-FIRST filtering approach against two others procedures classically applied on genes as testing units: (1) filter for variants with a functional impact expected to change the protein ("missense_variant", "missense_variant&splice_region_variant", "splice_acceptor_variant", "splice_donor_variant", "start_lost", "start_lost&splice_region_variant", "stop_gained", "stop_gained&splice_region_variant", "stop_lost", "stop_lost&splice_region_variant" and "stop_retained_variant"), and (2) filter on the MSC

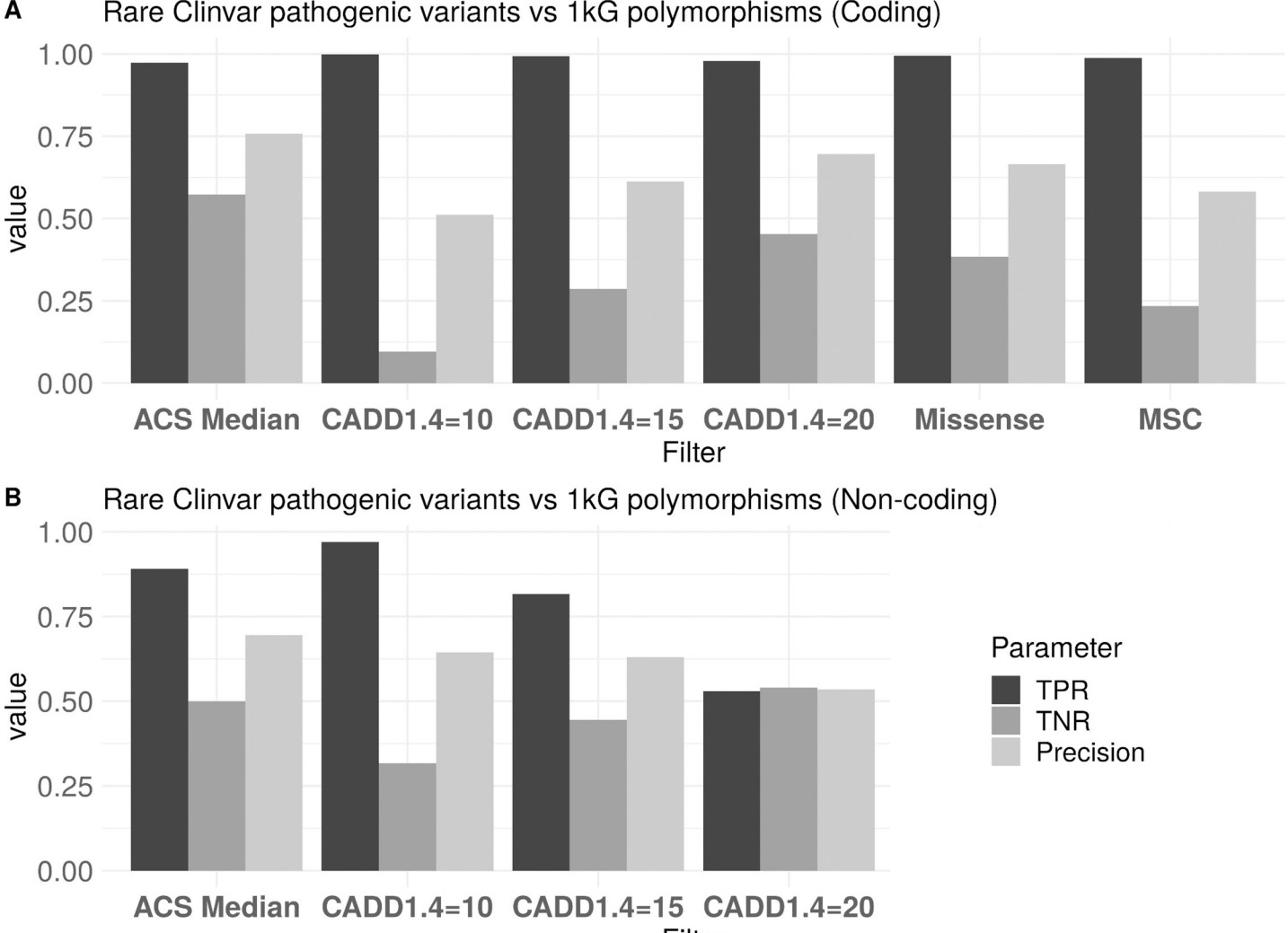

**Fig 2. TPR, TNR and precision of different filtering strategies on the Clinvar coding or non-coding variants pathogenic variants compared to rare 1000Genome polymorphisms.**

value, a gene-specific CADD threshold [30]. These two filtering approaches resulted in a slightly higher TPR than our proposed strategy but lower TNR and lower precision (Fig 2A). Therefore, even in an exome analysis, the RAVA-FIRST filtering would outperform classical filtering strategies to select qualifying rare variants for RVAT.

For non-coding variants, performances are lower than for coding variants. This is true when using both fixed CADD thresholds and the ACS median (Fig 2B) but the TPR is much lower when a fixed CADD threshold is used. This is explained by the fact that CADD values are lower in the non-coding genome. The best CADD threshold among hard-threshold filtering is indeed 10 in the non-coding genome while it is 20 in the coding genome. It is thus difficult to use a single fixed CADD value to select rare variants in testing units in the whole genome and the proposed ACS thresholds may therefore be preferred. Note however that, because of a bias towards coding variants in ClinVar pathogenic variants, the number of non-coding variants included in this analysis is rather low (2,980) compared to coding variants (79,831) and results should therefore be interpreted with caution.

In summary, compared to classical filtering strategies, the RAVA-FIRST approach based on ACS is expected to improve rare variant selection for RVAT in both the coding and the non-coding parts of the genome.

## RAVA-FIRST burden test–Simulations

To validate the RAVA-FIRST burden test, we performed simulations under the null hypothesis and under different scenarios of association using data from the 1000 Genomes European populations [29] in the *LCT* gene. We simulated 1,000 controls and 1,000 cases using the simulations based on haplotypes implemented in the R package Ravages [19]. A total of 201 variants was considered in the *LCT* gene. These variants were polymorphic in the European populations with a MAF lower than 1%. Two CADD regions overlap the *LCT* gene, R019233 and R019234, containing respectively 75 and 126 variants, both regions overlap coding and regulatory categories.

### Type I error

We first simulated data under the null hypothesis to verify that the RAVA-FIRST burden test maintains appropriate type I errors. We simulated two groups of 1,000 individuals in the R019234 CADD region without any genetic effect and we applied the classical WSS and the RAVA-FIRST WSS. Type I errors were computed using $5 \cdot 10^6$ simulations at three significance levels: $5 \cdot 10^{-2}$, $10^{-3}$ and $2.5 \cdot 10^{-6}$ (the usual threshold for whole exome rare variant association tests). The RAVA-FIRST WSS maintains good type I error levels at these different significance thresholds, similar to the ones obtained with the classical WSS (S3 Table).

### Power analysis

We then performed a power study with causal variants located either in the R019234 CADD region only or in the entire *LCT* gene in any of the two CADD regions. We simulated 50% of causal variants randomly spread in the whole unit (scenarios S1 and S3), in the coding regions (scenarios S2A and S4A) or in the regulatory regions (scenarios S2B and S4B). All the scenarios are summarised in Table 2. We compared the classical WSS to the RAVA-FIRST WSS using the gene or the two CADD regions as testing units. When CADD regions were used as testing units, analyses were performed for each of the two CADD regions and the minimum p-value was taken and multiplied by two to correct for multiple testing. A total of 1,000 replicates were simulated for each scenario and power was assessed at a genome-wide significance threshold of $2.5 \cdot 10^{-6}$.

**Table 2. Scenarios of association simulated to assess the performance of the RAVA-FIRST burden test.**

|  | *LCT* gene | | | |
|---|---|---|---|---|
|  | R019233 | | R019234 | |
|  | **Coding** | **Regulatory** | **Coding** | **Regulatory** |
| S1 |  |  | 50% | |
| S2A |  |  | 50% | 0% |
| S2B |  |  | 0% | 50% |
| S3 | 50% | | | |
| S4A | 50% | 0% | 50% | 0% |
| S4B | 0% | 50% | 0% | 50% |

**Table 3. Power at the genome-wide significance level of $2.5 \cdot 10^{-6}$ under the different simulation scenarios using either the classical WSS or the RAVA-FIRST WSS at the scale of either the entire gene or CADD regions.**

|  | By gene | | By CADD regions | |
|---|---|---|---|---|
|  | Classical WSS | RAVA-FIRST WSS | Classical WSS | RAVA-FIRST WSS |
| S1 | 0.409 | 0.370 | 0.782 | 0.701 |
| S2A | 0 | 0.431 | 0.002 | 0.602 |
| S2B | 0.408 | 0.404 | 0.689 | 0.706 |
| S3 | 0.751 | 0.678 | 0.512 | 0.433 |
| S4A | 0.004 | 0.564 | 0.012 | 0.474 |
| S4B | 0.657 | 0.64 | 0.39 | 0.391 |

Table 3 presents the power results obtained from this simulation study for both the classical WSS and the RAVA-FIRST WSS. Similar trends were observed between the two analyses, regardless if the simulations are performed at the scale of CADD regions or at the scale of the gene. When the causal variants were randomly sampled across the entire region (scenarios S1 and S3), the classical WSS with only one score for the entire region slightly outperformed the RAVA-FIRST method with sub-scores. Nevertheless, the loss of power for the latter was modest (less than 10%). By contrast, when causal variants were present only in the coding categories (scenarios S2A and S4A), which represent a small proportion of the entire region (approximately 15%), the RAVA-FIRST strategy was much more powerful than the classical WSS (approximately 50% gain in power). When causal variants were present in the regulatory categories only (scenarios S2B and S4B), both strategies showed similar power. All these results highlight the gain of power using the RAVA-FIRST WSS when a cluster of causal variants is present within a functional category of the CADD region while maintaining good power levels when causal variants are spread across the entire region. When comparing the simulations with causal variants sampled at the gene level or at the CADD region level, burden tests gathering variants within the corresponding testing units show, as expected, the highest levels of power. Nevertheless, the loss of power when using CADD regions as testing units instead of the entire gene is lower when causal variants are sampled across the entire gene (scenario S3) than the gain of power they present when causal variants are sampled within a specific CADD region (scenario S1). This is particularly true for the RAVA-FIRST WSS.

## Applications

### RAVA-FIRST analysis

RAVA-FIRST was used on whole-genome sequence (WGS) data from patients affected by venous thromboembolism (VTE). VTE is a multifactorial disease with a strong genetic component [31]. There exists a huge heterogeneity between patients in the age at first VTE event. To study the role of rare variants on VTE age of onset, WGS data were used from 200 individuals from the MARTHA cohort [32]. These individuals were selected among patients with unprovoked VTE event who were previously genotyped for a genome-wide association study [33] and present no known genetic predisposing factor. Individuals were dichotomized based on the age at first VTE event either before 50 years of age (early-onset) or after (late-onset). The threshold of 50 years was chosen based on the results of recent studies [34] that hint toward a genetic heterogeneity between these two groups. A quality control (QC) of the sequencing data was performed using the RAVAQ pipeline [35] (https://gitlab.com/gmarenne/ravaq). After QC, 184 individuals were included for analysis with 127 presenting an early-onset VTE and 57 a late-onset VTE. Only variants passing all QC steps and with a MAF lower than 1% in the

**Table 4. Number of testing units and variants kept under the three strategies.**

|  | Testing units | Filtering | Number of testing units | Number of variants |
|---|---|---|---|---|
| WGScan<br>Fixed CADD threshold | Sliding windows | MAF ≤ 1%<br>CADD v1.4 ≥ 15 | 377,092 | 96,347 |
| RAVA-FIRST units (CADD regions)<br>No CADD filtering | CADD regions | MAF ≤ 1% | 103,439 | 9,423,012 |
| RAVA-FIRST units (CADD regions)<br>Fixed CADD threshold |  | MAF ≤ 1%<br>CADD v1.4 ≥ 15 | 10,389 | 96,294 |
| RAVA-FIRST units (CADD regions)<br>RAVA-FIRST filtering |  | MAF ≤ 1%<br>ACS ≥ median | 95,220 | 3,494,327 |

sample were considered in the association tests comparing early and late-onset groups. For these comparisons, rare variants were gathered either by CADD regions or by using the sliding windows procedure implemented in WGScan [18]. Qualifying variants were selected based on CADD scores and using two filtering strategies: (1) a fixed CADD threshold of 15 or (2) the RAVA-FIRST CADD region-specific filtering (applied on ACS). Association was tested using the WSS burden test. When the RAVA-FIRST filtering was used, the corresponding WSS test with sub-scores was applied. Table 4 shows the number of testing units and variants kept under each strategy. For all tests with CADD regions, only testing units containing at least 5 rare variants were kept. WGScan was used with default parameters, i.e. with testing units of 5, 10, 15, 25 or 50 kb.

QQ-plots for the WSS tests using those three strategies are shown in Fig 3. As expected, a lower significance threshold is required to reach genome-wide significance with the sliding window procedure due to the higher number of testing units. Accordingly, the computation time was much lower for the two analyses by CADD regions (6min when filtering based on a fixed CADD score threshold and 25min when using the region-specific CADD thresholds) than for the sliding windows procedure (47min). Our dataset contains less than 200 individuals, suggesting that the gain in computation time of CADD regions compared to sliding window procedures would be even greater in larger WGS datasets. No significant result was found when no functional filter was applied nor when selecting variants with a CADD score greater than 15. One association reached borderline significance (p = $6.41 \cdot 10^{-7}$) when using the RAVA-FIRST strategy, i.e. with CADD regions and the corresponding ACS filtering.

This association maps to R126442, a CADD region of 21 kb on chromosome 18:66788277–66809402 that contains 30 rare variants after RAVA-FIRST filtering. In this region, none of the variants observed in VTE patients or in gnomAD achieved a CADD score above 15. This explains why the association could not have been detected by the two other strategies based on fixed CADD score ≥ 15. The median of ACS observed for gnomAD variants in this region is 1.73 and the ACS of selected variants range from 1.82 to 8.50. These observations emphasize the need to adapt thresholds depending on the genomic region under analysis. Interestingly, only early-onset VTE patients carry qualifying rare variants and have non-null WSS scores (Fig 4 and S3 File). Among early-onset patients, a trend is also observed for WSS scores to decrease with increasing age of onset. Information about the CADD region R126442 is available in the S3 File. Information about individuals (WSS score, age and sex) and variants (position, adjusted CADD score and weight in WSS) are given.

To make sure that there was indeed an advantage of using CADD regions compared to random windows over each chromosome, we shuffled the CADD regions on chromosome 18, computed new CADD medians in each region and tested for association again. We repeated this procedure 500 times and looked at the region where the top signal (lowest p-value) is located in each permutation (S3 Fig). We found an enrichment of top signals around R126442

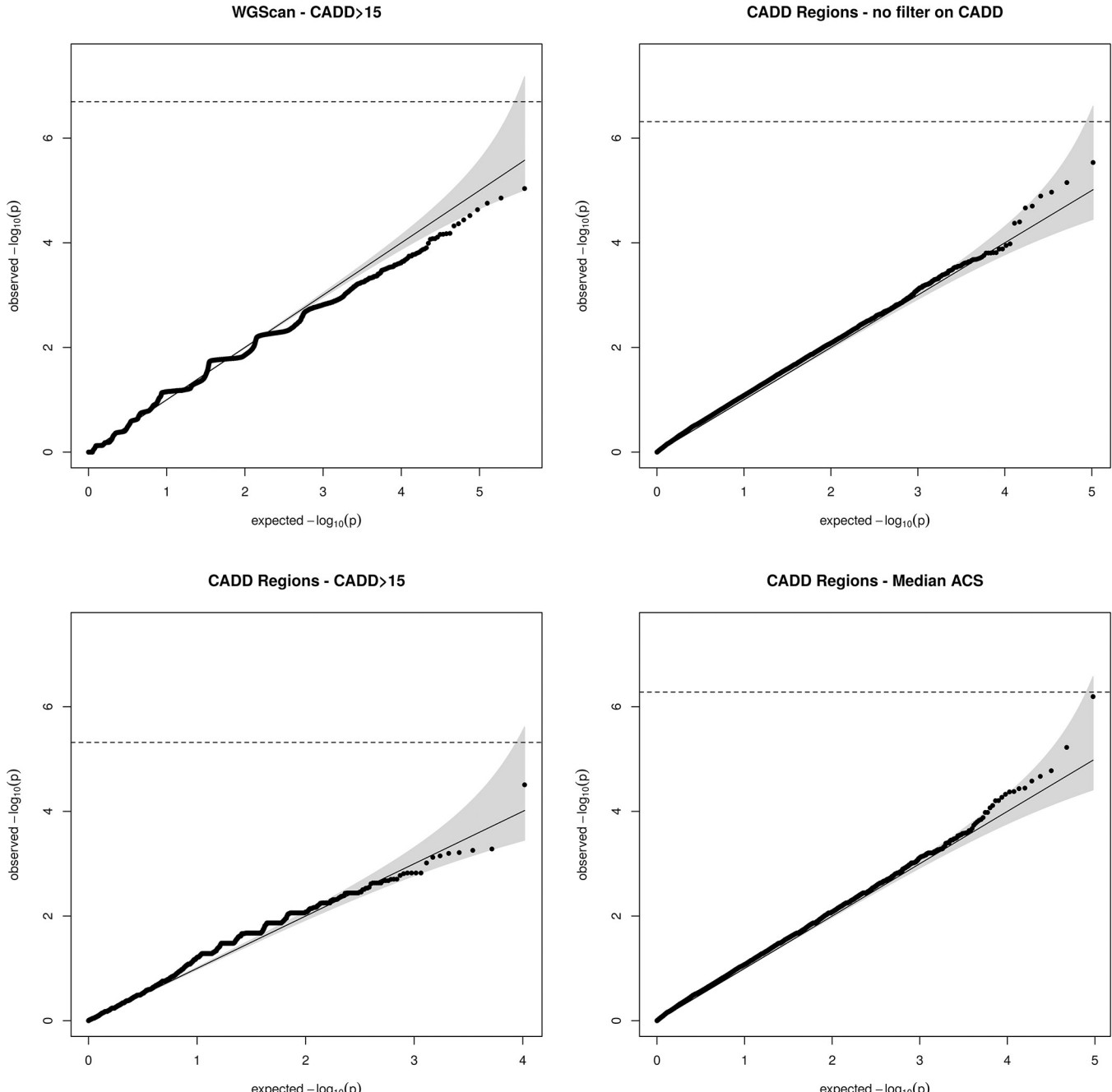

**Fig 3. QQ-plot of WSS analyses on VTE data using the four strategies of analysis.** Early-onset patients ($<$50 years old) were compared to late-onset patients ($\geq$50 years old).

and no other region in the chromosome reached the same significance level. Specifically, the top signal overlaps with R126442 in 34.2% of the replicates and this percentage increases to 62.4% if we consider the top 5 signals. The percentage of CADD regions overlapping with R126442 is yet smaller than 0.1% when looking at the whole p-value spectrum.

The CADD region R126442 was then tested for association with 20 biological VTE bio-markers available in MARTHA patients: antithrombin, basophil, eosinophil, Factor VIII,

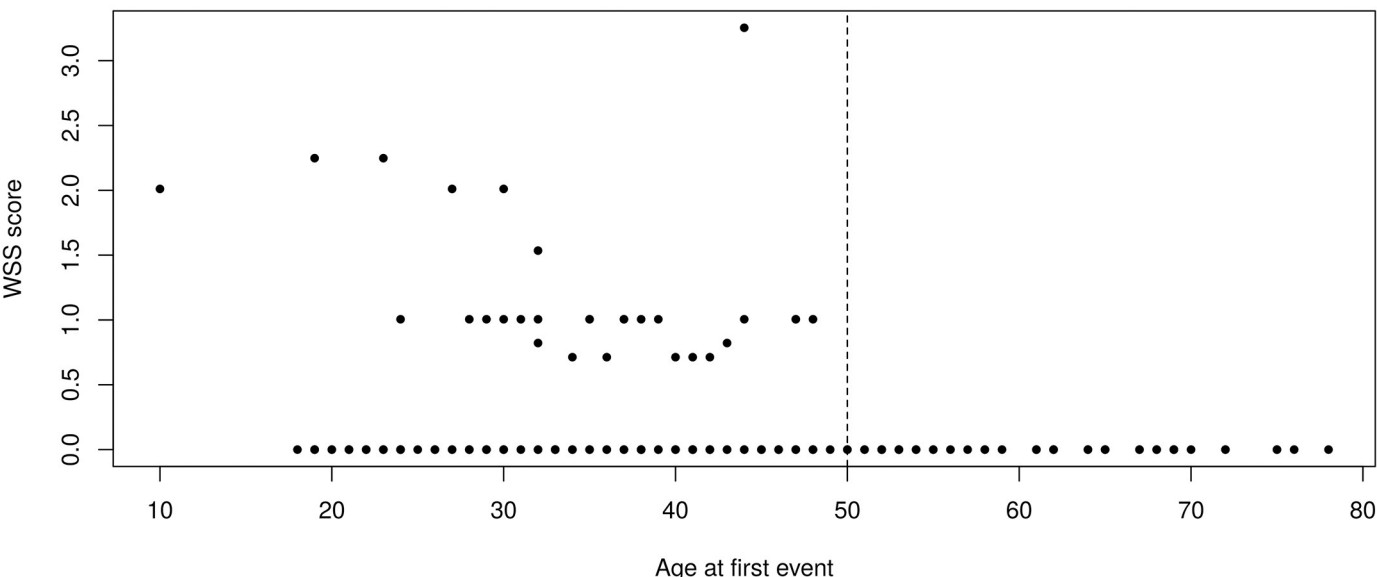

**Fig 4. WSS scores in the CADD region depending on the age at first VTE event.** The dashed line corresponds to the age 50 discriminating early onset from late onset events.

Factor XI, fibrinogen, hematocrit, lymphocytes, mean corpuscular volume, mean platelet volume, monocytes, neutrophils, PAI-1, platelets count, protein C, protein S, prothrombin time, red blood cells count, von Willebrand Factor, and white blood cells count. For this, a linear regression model was used where adjustment was made on age at sampling and sex. At the Bonferroni threshold of 0.0025, one significant association (p = $7.1 \cdot 10^{-4}$) was observed, VTE patients with a non-null WSS score exhibiting decreased haematocrit levels, a surrogate marker of red blood cells (S4 Table). A similar trend (p = $4.6 \cdot 10^{-3}$) was observed with red blood cell count.

We also investigated the association of the identified region with 376 plasma protein antibodies that were selected to be involved in thrombosis-related processes and that have been previously profiled in MARTHA [32,36]. Regression analysis were conducted on log transformed values of antibodies and were adjusted for age, sex, and three internal control antibodies. In order to handle the correlation between measured protein antibodies, we used the Li and Ji method [37] to estimate the number of effective independent tests. This number, calculated to be 163, was then used to define a Bonferroni threshold for declaring study-wide statistical significance. While not reaching the study-wise significance level of p = $3.1 \cdot 10^{-4}$ after correction for multiple testing, it is worth noting that the two proteins that exhibited the strongest significance with marginal association at p < 0.001, procalcitonin tagged by the HPA043700 antibody (p = $7.2 \cdot 10^{-4}$) and PDPK1 tagged by HPA035199 (p = $7.5 \cdot 10^{-4}$), were both suggested to be involved in red blood cell biology [38,39].

According to ENCODE data, the R1246442 CADD region overlaps "intergenic" and "regulatory" categories with one distant enhancer-like signature. To further describe this region, we looked at TADs positions in https://dna.cs.miami.edu/TADKB/brows.php in HUVEC and HMEC cell lines, two cell types known to be relevant for VTE pathophysiology. We found that the CADD region is included into the topological associated domain (TAD) 18:66450000–68150000. By studying TADs described by Lieberman-Aiden et al. 2009 [40] in other cell lines

such as KBM7, K562 or GM12878, we retrieved a TAD with similar positions, giving additional evidence for the presence of this TAD around the CADD region associated with early-onset patients. We then explored this TAD region for the presence of candidate VTE genes whose regulation could be influenced by the enhancer region that maps our R1246442 region. Using the UCSC genome browser [41] integrating information about interactions between GeneHancer regulatory elements and genes expression (see S4 Fig), we identified *CD226* as a strong biological candidate. *CD226* codes for a glycoprotein expressed at the surface of several types of cells, including blood cell, and several studies have shown that it was associated with vascular endothelial dysfunction [42–44]. Genetic variants in *CD226* have also been found associated with several blood cell traits including platelets, white blood cells (e.g. neutrophil, eosinophil) [45] and reticulocyte counts [46], another red blood cell biomarker.

## Discussion

Even though whole genome sequencing data are now widely available, rare variant association tests (RVAT) usually remain restricted to the coding parts of the genome. This is explained by the lack of tools to explore rare variant associations outside genes [11]. It is especially difficult to predict the functional consequence of non-coding variants and not currently possible to analyse them in RVAT without using computationally-intensive sliding window procedures. In this work, we propose RAVA-FIRST, an entire new strategy of analysis of rare variants in the coding and the non-coding genome that leverages functional information. RAVA-FIRST is composed of three steps. First, RAVA-FIRST groups variants observed in cases and controls into some new testing units, the so-called "CADD regions". These CADD regions are defined over the entire genome based on CADD scores of variants observed in gnomAD. They are large enough to include a sufficient number of rare variants to allow RVAT. They tend to preserve functional elements that, for a majority of them, are not split into several CADD regions. Second, RAVA-FIRST filters variants based on region-specific adjusted CADD thresholds that allow the selection of the best candidate variants within each region. This filtering approach was found to be more efficient than traditional approaches to discriminate between benign and pathogenic variants within a set of variants. Indeed, our benchmarking study using a set of Clinvar variants compared to 1000Genomes polymorphisms showed that the other filtering strategies were good at identifying true causal variants (true positive rates were high) but bad at finding the non-causal variants (true negative rates were low), especially in the coding genome. Both true positive and true negative rates are important to achieve a high percentage of causal variants within testing units, a major driver of power in RVAT, especially in burden tests [2,3,7]. Thus, the RAVA-FIRST filtering strategy is expected to result in an appreciable gain of power. Indeed, RAVA-FIRST enables to keep the most important functional variants within coding, regulatory and intergenic categories of the genome by adapting CADD score threshold to the genomic context. Third, RAVA-FIRST includes a burden test that integrates information on genomic categories in the regression model and that, coupled with the region-specific filtering, leads to a better detection of causal variants, should they cluster in one of these genomic categories only. We also showed through simulations that good power levels were maintained using RAVA-FIRST burden test when causal variants were randomly sampled.

RAVA-FIRST was applied on real WGS data from VTE patients where an accumulation of rare variants in patients with early-onset events was investigated. We did not detect any significant signal using the sliding window procedure or CADD regions when qualifying rare variants were selected based on a minor allele frequency threshold and/or a fixed CADD threshold. However, we detected an association signal using both the grouping and filtering of

rare variants proposed in RAVA-FIRST. The associated CADD region is intergenic, contains a predicted enhancer and is surrounded by a TAD containing 5 genes including *CD226*, a strong candidate for blood cell traits that are now well recognized to be key players in VTE physiopathology [31]. All rare variants in this region present low CADD scores and were not even included in analyses based on a fix CADD threshold, highlighting the importance of considering the genomic context to detect the most important predicted functional variants within each CADD region. These 30 rare variants are exclusively observed in early-onset cases. Fourteen of these variants are absent from gnomAD, and 10 of the 16 remaining variants have a lower frequency in gnomAD population than in our sample. This reinforces the value of the association signal in this CADD region, although it should be further described and validated using functional experiments. Preliminary investigations that need to be further explored, at both experimental and epidemiological levels, strongly suggest that this region is associated with several inflammatory markers impaired in anaemia of inflammation [39,47] and in platelets, both mechanisms being involved in thrombotic processes [48].

The RAVA-FIRST approach could be improved on different aspects. First, the definition of CADD regions relies on the gnomAD population and on the adjusted CADD threshold. We chose to use the whole gnomAD dataset but it could be of interest to select only some of the populations to detect population-specific associations that could for example be explained by ancestry-related differential expression patterns [49]. Nevertheless, in classical exome analyses, rare variants are usually filtered based on the maximum frequency observed among multiple populations. Furthermore, CADD regions are not defined for low-covered and non-sequenced genomic regions in gnomAD and their definition could benefit from the inclusion of data from other large population datasets where these regions are better covered. We also observed that CADD regions close to the centromeres can be very large, possibly due to less accurate annotation scores resulting from only few high quality genomes mapping these areas. We therefore recommend to cautiously interpret association signals that would be detected in these regions. To define the regulatory regions of the genome as one of the three genomic categories, we decided to include all genomic elements directly implicated into regulatory functions but we did not include silencers or lncRNA for example. However, the choice of elements to include as the regulatory category will only impact the adjusted CADD scores that are similar between regulatory and intergenic regions, and won't therefore have a huge impact on CADD regions definition. As an example, the use of DECRES [50] to predict enhancers and promoters instead of SCREEN results in a very high correlation between the definition of CADD regions, 80% of them being identical. The choice of focusing on variants seen at least twice in gnomAD and with an ACS larger than 20 could also be discussed. This decision was made based on a fine tuning to obtain testing units with sizes that were the most compatible with rare variant analysis, but this could also be adapted to the genomic context as we have done by grouping small regions where several variants showed high ACS.

By using CADD scores to define the testing units in RAVA-FIRST, we were able to propose a general framework to cover the entire genome. Indeed, while several other predictive tools have been proposed (as for example LINSIGHT [51], JARVIS [52] or ORION [53]), only few provide a score that is variant specific and defined in both the coding and non-coding parts of the genome. The use of the same framework to define testing units in the whole genome offers several advantages, including the region-specific filtering which enables to overcome the question of selecting a hard threshold to filter rare variants in RVAT. In addition, the newly defined CADD regions can be used in existing software that require regions as input parameters [54,55], enabling to apply a wide variety of RVAT available in those programs to the whole genome. Especially, Bayesian methods which have been shown to be of great promise in the analysis and filtering of rare variants [56,57] could be applied beyond genes by using CADD regions.

CADD regions represent predefined testing units for RVAT that cover the highest proportion of the genome and have been made publicly available. They are part of a whole new strategy of rare variant analysis in the whole genome, RAVA-FIRST, that further benefits from the integration of functional information both for the filtering of rare variants and their analysis with burden tests. RAVA-FIRST has been implemented in the R package Ravages available in the CRAN and on Github, offering an easy and straightforward tool to perform RVAT in the whole genome. We believe that our developments will help researchers to explore the role of genome-wide rare variants in complex diseases. Firstly, through the redefinition of testing units in the coding genome where cluster of causal variants can be found within genes and retrieved using CADD regions [10]. Secondly, through the study of non-coding variants, especially intergenic ones, which are currently often excluded from the analysis. Going beyond the gene and the consequences on proteins, RAVA-FIRST will help for a better understanding of biological mechanisms behind complex diseases.

## Supporting information

**S1 Fig. Definition of CADD regions and removal of low-covered and non-sequenced regions in gnomAD.**
(TIF)

**S2 Fig. Percentage of CADD regions overlapping each of the three genomic categories.**
(TIF)

**S3 Fig. Association analysis on VTE data on chromosome 18 by shuffling the CADD regions (500 replicates).**
(TIF)

**S4 Fig. Screenshot of the TAD 18:66450000–68150000 in the UCSC genome browser containing the CADD region R126442 and a potential enhancer regulating the CD226 gene, a candidate gene in VTE.**
(TIF)

**S1 Table. Sources used to get genomic elements for comparisons with CADD regions.**
(DOCX)

**S2 Table. Percentage of genomic elements entirely encompassed within a CADD region.**
(DOCX)

**S3 Table. Type I error of the classical WSS and the RAVA-FIRST WSS using $5 \cdot 10^6$ simulations under the null hypothesis.**
(DOCX)

**S4 Table. Characteristics of the studied VTE sample.** Mean (Standard Deviation) for quantitative variables. Count (%) for qualitative variables.
(DOCX)

**S1 File. Details about the RAVA-RIST method and its evaluation.**
(DOCX)

**S2 File. Variants used for the evaluation of RAVA-FIRST comparing ClinVar pathogenic variants to 1000Genomes polymorphisms.**
(XLSX)

**S3 File. Information on the CADD region R126442 associated with VTE age at onset.**
Information about individuals (WSS score, age and sex) and variants (position, adjusted

CADD score and weight in WSS) are given.
(XLSX)

## Author Contributions

**Conceptualization:** Ozvan Bocher, Hervé Perdry, Emmanuelle Génin.

**Formal analysis:** Ozvan Bocher, Marie-Sophie Oglobinsky, Suryakant Suryakant, David-Alexandre Trégouët.

**Funding acquisition:** Jean-François Deleuze, Pierre-Emmanuel Morange, David-Alexandre Trégouët, Emmanuelle Génin.

**Investigation:** Pierre-Emmanuel Morange.

**Methodology:** Ozvan Bocher.

**Project administration:** Pierre-Emmanuel Morange, David-Alexandre Trégouët.

**Resources:** Jean-François Deleuze, Jacob Odeberg.

**Software:** Ozvan Bocher, Thomas E. Ludwig, Gaëlle Marenne, Hervé Perdry.

**Supervision:** Emmanuelle Génin.

**Writing – original draft:** Ozvan Bocher, David-Alexandre Trégouët.

**Writing – review & editing:** Ozvan Bocher, Thomas E. Ludwig, Gaëlle Marenne, Pierre-Emmanuel Morange, David-Alexandre Trégouët, Hervé Perdry, Emmanuelle Génin.

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
