## [Decision Letter · Decision Letter 0]

5 Mar 2022

Dear Dr Bocher,

Thank you very much for submitting your Methods entitled 'Testing for association with rare variants in the coding and non-coding genome: RAVA-FIRST, a new approach based on CADD deleteriousness score' to PLOS Genetics.

The manuscript was fully evaluated at the editorial level and by independent peer reviewers. The reviewers appreciated the attention to an important problem, but raised some substantial concerns about the current manuscript. Based on the reviews, we will not be able to accept this version of the manuscript, but we would be willing to review a much-revised version.

We cannot, of course, promise publication at that time. From an editorial perspective, the reviewers, and in particular reviewer 2 raises several major issues. I do think these issues can be addressed, as this is a request for additional work rather than a fundamental challenge of the concept of the paper, but none of them is trivial. This means (i) benchmark the proposed methods against the most commonly used tools in the field, (ii) understand how variability in mutation rate can complicate data interpretation and (iii) include indels into the model (both reviewers made that point and this is a major source of rare deleterious variation). These combined additions represent a high bar, but methods for rare variant association testing are quite mature, which in turn raises the expectations for alternative approaches.

Should you decide to revise the manuscript for further consideration here, your revisions must address the specific points raised above. We will also require a detailed list of your responses to the review comments and a description of the changes you have made in the manuscript.

If you decide to revise the manuscript for further consideration at PLOS Genetics, please aim to resubmit within the next 60 days, unless it will take extra time to address the concerns of the reviewers, in which case we would appreciate an expected resubmission date by email to plosgenetics@plos.org.

[LINK]

We are sorry that we cannot be more positive about your manuscript at this stage. Please do not hesitate to contact us if you have any concerns or questions.

Yours sincerely,

Vincent Plagnol

Associate Editor

PLOS Genetics

David Balding

Section Editor: Methods

PLOS Genetics

Reviewer's C**omments to the Authors:**

Reviewer #1: In their publication 'Testing for association with rare variants in the coding and non-coding genome: RAVA-FIRST, a new approach based on CADD deleteriousness score', Bocher et. all describe a novel method for selecting disease-candidate variants among rare SNVs genome-wide. The described methodology is very interesting and enables new insights especially in previously less regarding genomic regions. I would recommend however to revise the text somewhat in order to improve readability:

major:

- I would recommend to define the adjusted CADD scores once and then stick to that. Maybe use an acronym (e.g. ACS (adjusted CADD score), RAVA-Score or something like that). In my eyes, the switches between adjusted and not-adjusted CADD scores (i.e. l. 154) make the manuscript hard to read, especially as you define the adjustment multiple times. Something similar may help to distinguish 'all' and '>1000bp' CADD regions

- Table 2: Are percentages only from within the chosen (>1000bp) CADD regions or all. Or why else are some coding CCDS not in a CADD region? Why are protein domains so much less covered (likelihood that entire domain is covered instead of % of each bp)?

- Fig 1: Why do you include missense and MSC in the top panel. I understand that those are the same as the bottom panel, but the empty plots should hence be excluded altogether. Maybe you could use that space on the upper panel to put (a single) legend for both panels there (i.e. the TPR, TNR and precision colours).

One should note that Figure 1 does not tell much as most variants in ClinVar are coding and thus the adjusted CADD score does not do much. You mostly just have a higher cut-off than CADD1.4=20 (similar S4, just lower) which decreases TPR and increases FPR

suggestion:

- I wonder what happens in the very large 'CADD regions' around the centromers where, presumedly, many variants are not scored at all and general genome conservation is low and hence the median, which may lead to small numbers of variants in slightly higher conserved areas to be considered significant, idk). I have no idea what effect this may have but I, personally, would probably implement a maximum length for CADD regions and split regions larger than, say, 200 kb into the maximum number so that each is at least 100 kb (just a suggestion, maybe this does not work as intended)

- Maybe it's just me, but I find Figures S1 (to a lesser degree S2) definitely more relevant for the main manuscript than any of the Tables. You are generally moving a lot of the method (i.e. definition of CADD regions) in the supplement that seems an important part of the manuscript

- l. 320/321 'as recommended by https://cadd.gs.washington.edu/info, version v1.4': not to be pedantic but I would interpret 'there is not a natural choice here -- it is always arbitrary' rather as a recommendation for a dynamic threshold like your method than a single fixed cut-off. Afterall, you are pretty much proposing a (automated) solution for a problem that has also been stated there

- l.156: ‘Note that because CADD scores are only available for SNVs’ -> CADD is available for InDel, consider carefully however if you want to include those in the analysis

minor:

- l. 89/90 'These regions prevent the use of sliding windows procedures while enabling the study of rare variants in the whole genome' -> I would use 'avoid' or 'replace' instead of 'prevent'

- l. 92 fix -> fixed

- (multiple) I assume 'package R Ravages' should be 'R package Ravages'

- (multiple) I would write 'gnomAD', not 'GnomAD'

Reviewer #2: In this manuscript, Bocher et al. attempt to define a new approach to performing rare variant burden tests in the non-coding genome. With whole-genome sequencing of large disease cohorts increasing at a rapid rate, identifying such methods is of value to the field. Unfortunately, the authors’ approach at defining functional units of the non-coding genome fails to account for major confounders. Furthermore, they have failed to benchmark against some of the most popular tools in the field as outlined in my comments below. In addition, their rationale for defining these CADD regions is very unclear to me.

1. The authors define the boundaries of their CADD regions as regions between two variants with an adjusted CADD score > 20. However, they do not consider mutation rate. In CCR, which the authors compare their approach to, Havrilla et al. cleverly used CpG density as a proxy for mutation rate and showed that this approach worked well. Unfortunately, because the CADD regions do not account for mutation rate, it is entirely unclear whether the lack of variation in a CADD region is due to selective pressure or lower mutability of that region due to decreased mutation rate.

2. I’m not convinced that the authors are detecting regions of the genome depleted for functional variation. I’d like to see how their regions compare to other methods that attempt to define constrained genomic regions (JARVIS, Orion, CDTS, LINSIGHT, etc.). The authors should compare performance of these approaches in classifying non-coding ClinVar variants. Also, the authors should not use ClinVar benign variants as a negative set in these comparisons, as most ClinVar benign variants are gnomAD polymorphisms. Because gnomAD variation was used to define this score, they run into confounding with this comparison. A better approach would be to use variants seen in other databases (e.g., DiscovEHR) but not gnomAD as a putative benign set.

3. In defining CADD regions, the authors required the variant to be seen in gnomAD > 2x, but they do not provide any rationale how they came to this threshold. Same goes for choosing a CADD threshold for the burden tests.

4. While the authors compare to sliding-window based burden approaches, they should also compare to how their approach compares to defining functional units using the many other intolerance methods mentioned above.

5. They exclude indels/structural variants because these variants do not have CADD scores. Doing so results in a tremendous loss of power: indels / SVs should have much larger effect sizes than SNVs in the non-coding genome. A burden model that includes SNVs with a CADD threshold + any indels / SVs should be more powered.

6. The authors find a suggestive association between a CADD region and VTE. I’d like to see how their approach performs without any CADD filter as a negative control. Furthermore, I would want to see a comparison that their CADD region-based burden analysis does better than randomly splitting the genome into random chunks (matched in size to the CADD regions).

**Have all data underlying the figures and results presented in the manuscript been provided?**

Reviewer #1: Yes

Reviewer #2: Yes

PLOS authors have the option to publish the peer review history of their article (what does this mean?). If published, this will include your full peer review and any attached files.

Reviewer #1: No

Reviewer #2: No

---

## [Decision Letter · Decision Letter 1]

18 Jul 2022

Dear Dr Bocher,

Thank you very much for submitting your Methods entitled 'Testing for association with rare variants in the coding and non-coding genome: RAVA-FIRST, a new approach based on CADD deleteriousness score' to PLOS Genetics.

We are happy to say that the substantial reviewers' comments have been addressed and we are therefore close to being able to accept the manuscript. However, reviewer 1 made some minor comments and we ask you to address these, after which we expect to formally accept the manuscript.

[LINK]

Yours sincerely,

Vincent Plagnol

Associate Editor

PLOS Genetics

David Balding

Section Editor: Methods

PLOS Genetics

Reviewer's **Comments to the Authors:**

Reviewer #1: All my previous concerns have been addressed. However, there are a few minor comments to the new sections that should be fixed prior to publication:

l. 196/197: "This is expected given the fact that conservation and thus CADD scores are low in these regions."

l. 462/463 "We also observed that CADD regions close to the centromeres can be very large, possibly due to a general low conservation in these areas"

> My understanding is that there are few high quality genomes that cover these areas (repetitive elements, GC content) and hence alignments are missing to generate proper conservation scores.

l. 239 "Fig 2B" -> that is 2A now, right?

l. 477 "if" -> "of" (or the sentence does not make sense)

l. 479 "use of a" -> "use of the"

optional:

Consider rereading and revising some sentences here and there. Quite a few feel unnecessarily lengthy.

E.g l. 478, you can remove "of them can" without changing context

Reviewer #2: The authors have addressed my concerns

**Have all data underlying the figures and results presented in the manuscript been provided?**

Reviewer #1: Yes

Reviewer #2: Yes

PLOS authors have the option to publish the peer review history of their article (what does this mean?). If published, this will include your full peer review and any attached files.

Reviewer #1: No

Reviewer #2: No

---

## [Editor Report · Decision Letter 2]

15 Aug 2022

Dear Dr Bocher,

We are pleased to inform you that your manuscript entitled "Testing for association with rare variants in the coding and non-coding genome: RAVA-FIRST, a new approach based on CADD deleteriousness score" has been editorially accepted for publication in PLOS Genetics. Congratulations!

Yours sincerely,

Vincent Plagnol

Academic Editor

PLOS Genetics

David Balding

Section Editor

PLOS Genetics

Comments from the reviewers (if applicable):

**Data Deposition**

http://datadryad.org/submit?journalID=pgenetics&manu=PGENETICS-D-21-01454R2

**Press Queries**

---

## [Editor Report · Acceptance letter]

12 Sep 2022

PGENETICS-D-21-01454R2 

Testing for association with rare variants in the coding and non-coding genome: RAVA-FIRST, a new approach based on CADD deleteriousness score 

Dear Dr Bocher, 

We are pleased to inform you that your manuscript entitled "Testing for association with rare variants in the coding and non-coding genome: RAVA-FIRST, a new approach based on CADD deleteriousness score" has been formally accepted for publication in PLOS Genetics! Your manuscript is now with our production department and you will be notified of the publication date in due course.

With kind regards,

Anita Estes

PLOS Genetics

On behalf of:
